# Engineered Dual Antioxidant Enzyme Complexes Targeting ICAM-1 on Brain Endothelium Reduce Brain Injury-Associated Neuroinflammation

**DOI:** 10.3390/bioengineering11030200

**Published:** 2024-02-21

**Authors:** Brian M. Leonard, Vladimir V. Shuvaev, Trent A. Bullock, Kalpani N. Udeni Galpayage Dona, Vladimir R. Muzykantov, Allison M. Andrews, Servio H. Ramirez

**Affiliations:** 1Department of Pathology & Laboratory Medicine, Lewis Katz School of Medicine at Temple University, Philadelphia, PA 19140, USA; brian.leonard@temple.edu (B.M.L.); trent.bullock@temple.edu (T.A.B.); andrews.allison@ufl.edu (A.M.A.); 2Department of Systems Pharmacology and Translational Therapeutics, Perelman School of Medicine, University of Pennsylvania, Philadelphia, PA 19104, USA; shuvaevv@pennmedicine.upenn.edu (V.V.S.); muzykant@pennmedicine.upenn.edu (V.R.M.); 3Department of Pathology, Immunology & Laboratory Medicine, University of Florida College of Medicine, Gainesville, FL 32610, USA; galpayagedonak@ufl.edu; 4Center for Substance Abuse Research, Lewis Katz School of Medicine at Temple University, Philadelphia, PA 19140, USA; 5Shriner’s Hospital for Children, Philadelphia, PA 19312, USA

**Keywords:** traumatic brain injury, neuroinflammation, brain vasculature, blood–brain barrier, nanomedicine, superoxide dismutase 1, catalase

## Abstract

The neuroinflammatory cascade triggered by traumatic brain injury (TBI) represents a clinically important point for therapeutic intervention. Neuroinflammation generates oxidative stress in the form of high-energy reactive oxygen and nitrogen species, which are key mediators of TBI pathology. The role of the blood–brain barrier (BBB) is essential for proper neuronal function and is vulnerable to oxidative stress. Results herein explore the notion that attenuating oxidative stress at the vasculature after TBI may result in improved BBB integrity and neuroprotection. Utilizing amino-chemistry, a biological construct (designated “dual conjugate” for short) was generated by covalently binding two antioxidant enzymes (superoxide dismutase 1 (SOD-1) and catalase (CAT)) to antibodies specific for ICAM-1. Bioengineering of the conjugate preserved its targeting and enzymatic functions, as evaluated by real-time bioenergetic measurements (via the Seahorse-XF platform), in brain endothelial cells exposed to increasing concentrations of hydrogen peroxide or a superoxide anion donor. Results showed that the dual conjugate effectively mitigated the mitochondrial stress due to oxidative damage. Furthermore, dual conjugate administration also improved BBB and endothelial protection under oxidative insult in an in vitro model of TBI utilizing a software-controlled stretching device that induces a 20% in mechanical strain on the endothelial cells. Additionally, the dual conjugate was also effective in reducing indices of neuroinflammation in a controlled cortical impact (CCI)-TBI animal model. Thus, these studies provide proof of concept that targeted dual antioxidant biologicals may offer a means to regulate oxidative stress-associated cellular damage during neurotrauma.

## 1. Introduction

The CDC estimates that at least 2.5 million individuals experience a TBI annually, resulting in 1.4 million emergency department visits, 235,000 hospitalizations and 52,000 deaths [1]. Importantly, these numbers do not include TBIs that are treated in outpatient settings or those that go unreported. TBI is particularly common in young adults and children, making it the leading cause of death and disability in the United States among these age groups [2,3]. The most recent data from the CDC report that children and adolescents had 16,070 TBI-caused hospitalizations and nearly 3000 TBI-related deaths [4]. To date, there are no effective pharmacologic treatments approved specifically for those exposed to a TBI. This leaves TBI patients to suffer the debilitating consequences of a head injury, potentially for the rest of their lives [5].

TBI pathology consists of primary and secondary phases, with primary injury occurring at the time of impact and secondary injury resulting from the biochemical, inflammatory, and structural responses to injury [6]. Hallmark features of TBI pathology include the disruption of the blood-brain barrier (BBB), increased oxidative stress, and neuroinflammation [7,8,9,10]. Left unchecked, these injurious processes spread from the focal site of impact into surrounding healthy CNS tissue, increasing the area of damage, similar to the concept of a watershed stroke. The dynamic and debilitating pathophysiology of TBI paired with the extremely limited treatment options make clear the need for therapeutic interventions that can help prevent the spread of injury.

By modulating the pathology of secondary injury, the chronic negative consequences of TBI may be minimized. The BBB regulates access of blood components and immune cells into the brain [11]. The role of the BBB is to maintain the delicate neuronal environment [11], and its dysfunction is observed in the pathology of neurotrauma, stroke, and neurological disease [12]. The injury induced by TBI damages cerebral blood vessels, which circumvents the barrier mechanisms of the BBB [6]. Once BBB breach occurs, secondary events, such as oxidative stress damage, are triggered that further contribute to injury [13,14]. Oxidative stress resulting from neuroinflammation and excitotoxicity may be used as a target for therapeutic intervention [14]. There are many sources of reactive oxygen species (ROS) in TBI, including the arachidonic acid pathway, mitochondrial damage, xanthine oxidase activity, and the oxidation of heme [15,16,17,18]. Of particular importance is the prevention of further mitochondrial damage. Dysfunction of this organelle not only decreases total cellular energy output and thus decreases regenerative capacity, but the mitochondria also create more ROS in a self-perpetuating and self-damaging cycle.

Immediately following a TBI, production of superoxide anions is massively increased by electron transport chain enzymes, xanthine oxidase, and NADPH oxidases from injured cells and from the recruited peripheral immune system [19,20,21]. High concentrations of superoxide overwhelm the endogenous antioxidant systems, allowing this anion to react to form additional oxidants that are more stable, such as peroxynitrite or a highly reactive hydroxyl radical via hydrogen peroxide and free heme from damaged red blood cells. These free radicals have deleterious effects on cell physiology, disrupting cell membranes through lipid peroxidation and oxidizing cellular proteins and DNA, which in turn creates more superoxide. This damaging cycle is caused by the cells redox balance, being strongly pro-oxidant and not counteracted by the actions of limited and depleted amounts of endogenous antioxidant enzymes.

In order to correct against the damaging wave of oxidative stress causing imbalance, the notion of targeting two antioxidant enzymes (SOD-1 and catalase) conjugated with an anti-ICAM1 antibody was explored in the studies herein. Early attempts at this strategy have shown strong results. The administration of anti-ICAM-1 antibodies conjugated with catalase following moderate TBI showed brain endothelial cell protection, tissue sparing, reduced inflammatory response, and long-term functional improvement [22,23]. Adding SOD-1 to the anti-ICAM-1–catalase conjugate brings two natural antioxidant enzymes directly to the damaged BBB endothelium, where they can dismutate superoxide into hydrogen peroxide (SOD-1) and then immediately convert that hydrogen peroxide into water and oxygen (catalase). Targeted dual antioxidant enzyme therapy could thus help restore the redox balance within damaged cells by decreasing the free radical burden brought on by TBI. The results presented in this report provide proof of concept that the concerted actions of a catalase and superoxide dismutase conjugate cross-linked with an anti-ICAM-1 antibody into nanoparticles greatly lessens the neuroinflammation associated with TBI. Specifically, this study provides evidence that addition of this dual conjugate (1) greatly attenuates the oxidative stress imbalance during neurotrauma (2) preserves BBB integrity and function and (3) reduces inflammatory signatures in the CNS parenchyma.

## 2. Materials and Methods

### 2.1. Dual-Conjugate Preparation

Antioxidant enzymes were acquired from commercial sources. Bovine liver purified catalase was obtained from Calbiochem (San Diego, CA, USA), and superoxide dismutase 1 (SOD-1) from bovine erythrocytes was purchased from EMD Chemicals (San Diego, CA, USA). Succinimidyl-6-[biotinamido]hexanoate (NHS-LC-biotin), 4-[N-maleimidomethyl]cyclohexane-1-carboxylate (SMCC), and N-succinimidyl-S-acetylthioacetate (SATA) were purchased from Thermo Fisher Scientific (Waltham, MA, USA). Clone YN1/1.7.4 is a monoclonal antibody of the IgG2b isotype (mAb) against anti-intercellular adhesion molecule 1 (ICAM-1) [24].

Antibodies were conjugated with catalase and SOD-1 using amino-chemistry fusion methods as previously described [23]. Protected SH groups were introduced in the molecule of 5.2 mg/mL antibody via primary amines using SATA at an initial molar ratio of 1:10 of antibody:SATA at room temperature for 30 min followed by SH group deprotection with 50 mM hydroxylamine for 2 h. Maleimide groups were introduced into 12 mg/mL catalase and 3.0 mg/mL SOD-1 using SMCC (SOD:SMCC 1:8 and catalase:SMCC 1:20 initial molar ration) for 1 h at room temperature. After protein modification, unreacted reagents were removed by 7K MWCO Zeba Spin Desalting Columns (Thermo Fisher Scientific; Waltham, MA, USA). Conjugation was performed at a 2:1:1 Ab:Catalase:SOD molar ratio on ice and stopped when particle size reached 250–300 nm, as measured by a dynamic light scattering (DLS) Zetasizer Nano ZSP (Malvern Instruments Ltd., Malvern, UK). Conjugates were resuspended with sucrose as a cryoprotector at a final concentration of 20%, then frozen and stored at −80 °C before use. Conjugation was further confirmed by denaturing non-reducing 4–15% SDS-PAGE in a tris-glycine system. The control mixture contained equivalent amounts of unmodified antibody, SOD1 and catalase.

### 2.2. Cell Culture and Reagents

All experiments used the bEnd.3 cell line, an immortalized brain endothelial cell line derived from mouse cortex and acquired at low passage from the ATCC (Manassas, VA, USA). Cells were grown on rat-tail collagen I-coated flasks in Dulbecco’s modified Eagle’s medium (DMEM) supplemented with 10% fetal bovine serum, 1% Fungizone, and 1% penicillin–streptomycin (Thermo Fisher Scientific, Waltham, MA, USA). bEnd.3 cells were grown in an incubator set to 37 °C, 5% CO_2_, and 95% humidity.

Hydrogen peroxide (H_2_O_2_) at 30% *w*/*w* solution in water with a stabilizer was purchased from Sigma-Aldrich (St. Louis, MO, USA) and diluted to the desired molarity using serum-free DMEM. For all in vitro experiments, dual conjugates and controls were also diluted in serum-free DMEM prior to use.

### 2.3. Paracellular Permeability Assays

In order to examine bEnd.3 paracellular permeability, cells were seeded at a density of 16,000 cell per collagen-I coated Transwell inserts (Corning Life Science, Glendale, AZ, USA) on 24-well plates. Pore sizes were 0.4 μm, with a diameter of 0.33 cm^2^. Cells were seeded in 200 μL of DMEM with 10% FBS, and the basolateral chambers (surrounding the Transwell inserts) were filled with 500 μL of media. Media were changed every 2 days until confluent cell monolayers were formed (~5–6 days). Upon formation of a bEnd.3 cell monolayer, the cells were serum-starved overnight and then treated with 500 μM H_2_O_2_, 100 ng/mL dual conjugate, or both in the treatment group for 1 h. Prewarmed and well-mixed FITC–dextran (3 kDa MW, Thermo Fisher Scientific, Waltham, MA, USA) was then added to the inserts at a resulting concentration of 2 mg/mL. After 1 and 3 h, media were taken from the basolateral chambers and the fluorescence of the FITC–dextran permeability tracer was measured at 525 nm using a SpectraMax M5e (Molecular Devices Inc., San Jose, CA, USA).

### 2.4. Metabolic Flux Assays

Cells were seeded at a density of 16,000 per well onto a Seahorse XFe96 Culture Microplate (Agilent Technologies Inc., Santa Clara, CA, USA) in DMEM with 10% FBS. After seeding, plates were allowed to rest in the culture hood for 30–45 min to reduce edge effects prior to placement in an incubator overnight. The Seahorse XFe96 sensor cartridge was prepared with cell culture-grade sterile water (Corning, Corning, NY, USA) and placed into a 37 °C non-CO_2_ incubator along with Seahorse Calibrant solution. The following day, the hydrated cartridge was switched to calibrant and placed back into the non-CO_2_ incubator. Seahorse DMEM assay medium was prepared by supplementing it with pyruvate, glutamine, and glucose to mimic the DMEM the cells were grown in; phenol red and serum were not included in the seahorse assay medium. The complete Seahorse DMEM assay medium was then warmed in a 37 °C water bath until needed. H_2_O_2_ was added to wells for a final concentration of 500 μM and left for 1 h in the incubator. After the hour, either dual-conjugate or control groups (unconjugated mixture, free enzymes (SOD-1–catalase) and anti-ICAM-1 antibody) were added to a final concentration of 100 ng/mL in each well.

The Seahorse XF Cell Mito Stress Test Kit was prepared according to Agilent instructions and reagents were diluted in seahorse DMEM. Final concentrations injected into each well during the assay were as follows: oligomycin at 1.5 μM (mixture of oligomycin A, B, C with A ≥ 60%), carbonyl cyanide-4 [trifluoromethoxy] phenylhydrazone (FCCP) at 0.5 μM, and a mixture of rotenone–antimycin A at 0.5 μM. Oligomycin, FCCP, and rotenone–antimycin were then very carefully loaded in the Seahorse XF cartridge into ports A, B, and C respectively. After loading, the cartridge was placed back into the non-CO_2_ incubator until the assay was ready to start.

After the 2 h H_2_O_2_/drug exposure window, standard DMEM without serum was then carefully removed from the cell microplate media and the cells were washed twice with Seahorse assay medium. After the final wash, assay medium was added to the wells and the microplate was placed into the non-CO_2_ incubator for 45 min to degas any CO_2_ in the wells. The cartridge was added to the Seahorse XFe96 first to allow for calibration and then the microplate with cells was placed into the instrument. The standard Mito Stress Test assay parameters were initiated in Seahorse Wave Controller 2.6.3.8 software (Agilent) and the assay was run as per the manufacturer’s instructions [25].

### 2.5. Mechanical Stretch Injury to Cells (In Vitro TBI Model)

A Cytostretcher mechanical stretch device (Curi Bio Inc., Seattle, WA, USA) with stretch flexible cell culture chambers was used to generate mechanical strain. Membranes were washed with cell culture-grade sterile water, autoclaved and then plasma treated at 100 mTorr for 5 min prior to use. After plasma treatment, the membranes were collagen I-coated overnight. Cells were seeded at a density of 75,000 per Cytostretcher chamber (144 mm^2^, unpatterned “flat” surface topography membranes) in 10% FBS DMEM media. Medium was changed daily until cells formed a confluent monolayer (~6 days). After monolayers were formed, the cells were switched to serum-free media. The Cytostretcher device was housed within the incubator with the internal region of the device (where the membranes are housed and stretched) and maintained at 37 °C and 95% humidity. Using the NanoSurface Operational Mechanics Interface (NaOMI) computer software (Curi Bio, Seattle, WA, USA Version 4.1.8), stretch parameters were programmed in. All membranes tested were stretched by 4 mm, which is equivalent to a 20% strain on the cell monolayer. Additionally, there was a 0.5 s dwell time after stretching and before compression, thereby mimicking the biomechanical forces commonly observed in TBI acceleration/deacceleration-associated injury. After the cell-containing membrane chambers were stretched, they were immediately exposed to dual-conjugates (100 ng/mL final concentration) and controls (100 ng/mL final concentration), or vehicle.

After the treatment period, media were removed and cells washed with 1× PBS twice. Following the second wash, Accutase (Sigma Aldrich, St. Louis, MO, USA) was added to each membrane to lift the cells for further analysis. Cells were spun at 1000 RPM for 5 min and then resuspended with 10% FBS DMEM. Cells were then plated onto Seahorse microplates at 16,000 cells per well and incubated for 1–2 days for later bioenergetic assays, as outlined above.

### 2.6. Vertebrate Animals and Controlled Cortical Impact (CCI) Traumatic Brain Injury Model

The Institutional Animal Care and Use Committee (IACUC) at Temple University (Philadelphia, PA, USA) approved all procedures detailed in this study that required the use of vertebrate animals prior to initiating any experimental objectives. Additionally, all methods were performed in full compliance with Temple University’s IACUC policies and the National Institutes of Health (NIH) ethical guidelines. Six-week-old male C57BL/6 mice were purchased from the Jackson Laboratory (Bar Harbor, ME, USA). Animals were housed in cages and allowed to acclimate for two weeks in the Temple University Central Animal Facility. The animals were provided standard environmental enrichment conditions and were fed a commercial pellet diet and water ad libitum. Animals were weighed daily before and after surgery to ensure a stable body weight prior to euthanasia. The mice were anesthetized using 5% (induction) and 2% (maintenance) isoflurane in oxygen; consciousness was checked by hind paw pinching throughout the surgery. Surgical procedures for the experimental TBI model were followed as previously described [23].

Briefly, a 4 mm craniectomy was performed over the right somatosensory cortex between bregma and lambda suture lines. Artificial cerebrospinal fluid was dropped onto the surgical area as needed to prevent drying out and overheating from drilling. A moderate controlled cortical impact (CCI) was delivered at a speed of 3.5 m/s with a compression depth of 1 mm using an Impact One™ Stereotaxic CCI Instrument (Leica Microsystems; Buffalo Grove, IL, USA) outfitted with a 2 mm-diameter piston. The dwell time was 0.5 s. After impact, the craniectomy was sealed with a 5 mm glass coverslip (Electron Microscopy Sciences; Hatfield, PA, USA) using VetBond to allow for monitoring of the impact site.

All animals were individually housed following the CCI-TBI procedure. Naïve animals did not undergo any surgical intervention. Dual conjugates (1.0 mg/kg of anti-ICAM-1–SOD-1–catalase) or an unconjugated mixture of anti-ICAM-1, SOD-1, and catalase (1.0 mg/kg) were administered by retro-orbital injection 4 h following impact. Experimental groups were as follows: non-craniotomized controls (naïve), CCI-TBI only, CCI-TBI + dual conjugates, or CCI-TBI + unconjugated mixture.

### 2.7. Immunohistochemistry

Immunohistochemistry was performed on brain tissue segments to evaluate the extent of neuroinflammation and neuropathology in all experimental groups listed above. At 48 h after CCI-TBI, mice were anesthetized and then transcardially perfused with PBS followed by Poly/LEM fixative (Polysciences, Inc., Warrington, PA, USA). Perfused brains were removed from the skull and placed in Poly/LEM fixative for 24 h at 4 °C. Brains were then dissected into 2 mm segments using a stainless-steel brain matrix (CellPoint Scientific, Inc.; Gaithersburg, MD, USA). Segments were post-fixed in Poly/LEM fixative at 4 °C for an additional 24 h. Next, segments were washed with PBS, processed using a Tissue-Tek VIP6 (Sakura Finetek USA, Inc., Torrance, CA, USA), paraffin-embedded using a TN-1500 Embedding Console System (Tanner Scientific, Inc., Sarasota, FL, USA), and sectioned using a rotary microtome (Leica Microsystems, Inc., Buffalo Grove, IL, USA). Paraffin-embedded sections (5 μm) from each experimental group were cleared, rehydrated and stained with antibodies against NeuN, GFAP or Iba-1 to determine neuronal survival, extent of astrogliosis, or microglial activation. respectively. in the region of impact. Prior to primary antibody incubation. Sections stained for GFAP and Iba-1 were HIER-pretreated with 10 mM citric acid buffer (pH 6.0). Sections stained for NeuN received no pretreatment. All sections were incubated in primary antibody prepared in Dako Antibody Diluent either for 1 h at RT (NeuN, Iba1) or overnight at 4 °C (GFAP) at the following dilutions: NeuN (1:500, Abnova Corp., Walnut, CA, USA), Iba-1 (1:400, (Fujifilm Wako Chemicals Inc., Richmond, VA, USA), GFAP (1:2000, Cell Signaling Technologies Inc., Danvers MA, USA). Positive antibody staining was detected using an HRP- or AP-conjugated labeled polymer system (ImmPRESS Staining Kits, Vector Laboratories, Newark, CA, USA) and subsequently visualized using Sigma DAB (NeuN and Iba1) or Vector Blue (GFAP). Sections were then dehydrated, and cover-slipped in preparation for imaging.

### 2.8. Reactive Oxygen Species (ROS) Detection Assay

Cells were seeded at a density of 55,000 per well on a 24-well cell culture plate and allowed to grow until confluent monolayers were formed. Prior to exposure, wells were changed to 0% FBS DMEM media and cells were given time to adjust. MitoSOX red (Thermo Fisher Scientific, Waltham, MA, USA) was prepared in sterile, anhydrous DMSO to make a stock solution of 5 mM as per manufacturer instructions. A working solution was then prepared in DPBS with calcium and magnesium to yield a 500 nM working solution (protected from light at all times). Cells were exposed to either H_2_O_2_, SIN-1, and/or 1 mM KO_2_ (superoxide anion solution), as indicated in the Results. The superoxide anion solution was prepared as described previously [26,27]: 60 mg of 18-crown-6 ether was made into a fine powder and immediately dissolved in 10 mL of sterile, anhydrous DMSO, and 7 mg of powdered KO_2_ was quickly added to the solution to prevent any reactions with water vapor in the air. This mixture was then heated in a water bath for 10 min followed by vigorous mixing on a shaker for 1 h until a pale-yellow solution was formed with no remaining crystals of KO_2_. Appropriate concentrations of the mixtures were added to their respective wells and incubated for 1 h. Next, 100 ng/mL of dual conjugate or the unconjugated mixture were added, and the plates were once again incubated for 1 h. Cells were then washed with media to remove reagents and drugs, and then 1 mL of the MitoSOX red working solution was added to each well. Plates were then covered and incubated for 45 min, after which they were washed 3×with warm DPBS with calcium and magnesium. Cells were imaged using an EVOS FL Cell Imaging System (Thermo Fisher Scientific, Waltham, MA, USA). In order to quantify the fluorescent data from the above experiments, 96-well black/clear-bottom plates were used and the wells were read on a plate reader (SpectraMax M5e, Molecular Devices, San Jose, CA, USA) using the spectral properties provided by the manufacturer.

### 2.9. Immunofluorescence Assay

Immunofluorescence detection of ZO-1 was performed as previously described [28]. Briefly, cell monolayers were washed with 1× PBS and fixed with 4% paraformaldehyde for 10 min. Cells were then washed with 1× PBS, blocked and permeabilized with 5% donkey serum and 0.3% Triton-X for 20 min. To visualize the tight-junction protein zonula occludens 1 (ZO-1), cells were stained anti-ZO-1 (BD Biosciences, Franklin Lakes, NJ, USA) and diluted at 1:100 for one hour at RT. Secondary antibody anti-mouse Alexa Fluor 488 (Thermo Fisher Scientific, Waltham, MA, USA) was used at a dilution of 1:500 for 1 h at RT. The cell nuclei were counterstained with DAPI and the monolayer was mounted with ProLong antifade reagent. Images were acquired using a Nikon A1R inverted confocal microscope (Nikon Instruments, Tokyo, Japan).

### 2.10. Imaging and Image Analysis

Analysis of chromogen immunostaining of GFAP, Iba-1, and NeuN was performed with FUJI/ImageJ analysis software (Version 1.54g; NIH, Bethesda, MD, USA) using manual counting or particle counting on high-resolution images taken with a Nikon 80i microscope (Nikon instruments, Tokyo, Japan; 5/sample, 3/group, taken at 20× objective magnification) on immunolabeled cells. For manual counting, DAB+ nuclear staining in sections immunolabeled for NeuN were counted in an area of 2.5 × 10^5^ microns^2^. The hematoxylin counterstains nuclei to aid classification of DAB+ NeuN live or dead neurons. GFAP manual counting for activated astrocytes was performed based on density/color intensity above twice the threshold of that found in the naïve controls. DAB+ IHC for IBA-1 was counted as positive based on resting or activated microglia based on classical morphological presentation of their cellular status. For automated particle counting, the images were calibrated and then sequentially processed for background subtraction, color threshold segmentation, and binary conversion (with application of the watershed function) based on the above indicated parameters for manual counting. Cells identified from the above parameters were counted in the entire area of the image by use of the “analyze particles” function (selected by particle area and circularity).

Assessment of dual-conjugate cellular binding and internalization was performed by indirect immunofluorescence and confocal imaging. bEnd.3 cells were grown to confluence and then stimulated with 50 ng/mL of LPS (Sigma, St. Louis, MO, USA) overnight. Cells were then exposed to 100 ng/mL of dual conjugate. Cells were fixed/permeabilized in 4% formaldehyde for 15 min. The fixative was rinsed and permeabilization solution consisting of 0.1% Triton X-100 in 1× PBS was added to the cells for 10 min. Cells were placed in blocking solution (1% donkey serum in 1× PBS) for 1 h and then rinsed 3× with 1× PBS. The primary antibodies anti-ICAM-1 (1:200, Abcam, Waltham, MA, USA), anti-SOD-1 (clone 2F5, 1:200, Thermo Fisher Scientific, Waltham, MA, USA) were diluted in blocking solution. Primary antibodies were incubated with the cells overnight at 4 °C. Cells were then rinsed 3× in 1× PBS and secondary antibodies (Thermo Fisher Scientific, Waltham, MA, USA) were added for 2 h at room temperature. Secondary antibodies used were anti-rabbit Alexa Fluor 488 and anti-mouse Alexa Fluor 594. Cells were then rinsed 3× with 1× PBS and incubated with the nuclear stain counterstain DAPI (4′,6-diamidino-2-phenylindole, Invitrogen/Thermo Fischer Waltham, MA, USA), then rinsed with 1× PBS, and mounted using ProLong antifade reagent (Thermo Fisher Scientific, Waltham, MA, USA). Immunofluorescence was visualized using a Nikon A1R confocal microscope (Nikon Instruments, Tokyo, Japan). Images were acquired with Nikon NIS Elements AR imaging software (Version AR 2.2; Nikon, Tokyo, Japan).

### 2.11. Statistical Analysis

For in vivo experiments, all animals were randomly assigned to an injury group and endpoint time (n = 3). Tissue was isolated from both the contralateral and ipsilateral hemispheres to the craniectomy and the corresponding anatomical location in naïve mice. Data from image analysis were compared using one-way ANOVA and Dunnett’s post hoc test for multiple comparisons against the naïve control. All data are expressed as means ± S.E.M. For all in vitro cellular and biochemical assays, the experiments were conducted at least three times (three to four replicates per experiment) and compared using one-way ANOVA with post hoc Tukey’s or Dunnett’s test, as indicated in the Results section. Results are expressed as means ± S.E.M. All data were analyzed with GraphPad Prism software v9.5.1 (GraphPad, San Diego, CA, USA).

## 3. Results

### 3.1. Dual Antioxidant Enzymes and Targeting Anti-ICAM-1 Ab Mitigate Mitochondrial Stress Due to H_2_O_2_

Traumatic brain injury is known to produce reactive oxygen species that contribute to secondary damage after the initial injury. We have also previously shown the benefits of conjugates comprised of catalase connected to ICAM-1 antibodies [23]. This approach successfully reduced the pathophysiology after a TBI. Catalase disproportionately degrades hydrogen peroxide. In order to potentiate a further ROS degradation, we hypothesized that a combination approach that utilizes two enzymes (catalase and SOD) would advance this therapeutic technology. SOD targets superoxide radicals, converting them into hydrogen peroxide. Thus, the combined approach converts the main toxic species (superoxide radical and hydrogen peroxide) [29]. A new conjugate construct was engineered by first modifying mouse reactive anti-ICAM-1 antibodies (YN1/1.7.4) using the thiolating reagent N-succinimidyl S-acetylthioacetate (SATA). This process adds protected sulfhydryls that can then be used to link the antibody to additional molecules. Catalase and SOD-1 enzymes were also modified using the cross-linker sulfosulfosccinimidyl-4-(N-maleimidomethyl)-cyclohexane-1-carboxylate (sulfo-SMCC). Once the sulfhydryls are deprotected, the interaction between the antibody and enzymes chemically links the molecules, producing an enzyme–antibody complex (Figure 1).

To validate that the enzymes within the conjugate remained catalytically active, cells were exposed to either hydrogen peroxide (H_2_O_2_) or 3-morpholino-sydnonimine (SIN-1). Free radicals such as superoxide, hydrogen peroxide, and peroxynitrite are all present at the brain endothelium in TBI, initially localized in the impacted areas and later as a function of the diffused inflammation. The evolving ROS response shifts the cellular redox status to a pathological state that can leave cells permanently damaged or destroyed [8,18,30]. Furthermore, TBI due to hypoxia and compromised blood flow leaves injured cells in a highly glycolytic and anaerobic state [31,32]. Therefore, evaluation of cellular bioenergetics measured in real time via cell metabolic assays provides an assessment of both conjugate enzymatic function and efficacy to reduce oxidative stress response. Using the mouse brain endothelial cell line bEnd.3, cells were exposed to 500 μM of H_2_O_2_ (Figure 2), as these were empirically determined concentrations that induced lasting damage to cells that was only recoverable through treatments. Firstly, ATP production and its source were assessed using the Agilent ATP Rate Assay to determine the degree of cellular oxidative stress response as a function of H_2_O_2_ exposure. As Figure 2A demonstrates, addition of 500 μM of H_2_O_2_ to cells resulted in a doubling in ATP produced via glycolysis with a concurrent reduction in ATP by nearly threefold produced from the mitochondria compared to vehicle-treated cells. Additionally, the efficacy of the anti-ICAM-1–SOD-1–catalase dual conjugate was tested in a dose-dependent manner as well as against the specific control groups based on the individual components present in the conjugate. Cells in the presence of the dual conjugate enabled mitigation of the stressors that protected and promoted recovery of mitochondrial ATP production in a concentration-dependent fashion (Figure 2B). Cells treated with 100 ng/mL of dual conjugate started at a significantly higher baseline of oxygen metabolism than either the 50 ng/mL group (*p* = 0.01) or 10 ng/mL group (*p* = 0.001). During the FCCP portion of the assay, the 100 ng/mL group reached higher average OCR values than the 50 ng/mL group, but this was not statistically significant. However, comparing the maximal OCR values attained compared to the baseline OCR reveals the untreated group increased 31%, the 100 ng/mL group increased 23%, 50 ng/mL increased 8%, and the 10 ng/mL increased by 1%. There was no statistically significant difference between the 50 ng/mL and 10 ng/mL groups during this assay. Cells were also treated in the same manner as above, but exposed to the same concentrations of either anti-ICAM-1–SOD-1–catalase dual conjugate, SOD-1 and catalase as free enzymes, or SOD-1, catalase, and anti-ICAM-1 in an unconjugated mixture. As observed in Figure 2C, the addition of the dual conjugate outperformed the unconjugated mixtures in protecting against respiratory compromise from H_2_O_2_. The results suggest that the dual conjugate allows for the therapeutic antioxidant enzymes to be brought into proximity to the cell membrane and maintain a proper redox status in this microenvironment. In addition to testing the dual conjugates against a toxic insult such as hydrogen peroxide, further experiments revealed that the dual conjugates themselves had no deleterious effects on brain endothelial cells. Figure 2D,E shows that when bEnd.3 monolayers were exposed to the dual-conjugate or control groups, there was no difference in the OCR or ECAR when compared to the untreated cells.

### 3.2. Anti-ICAM-1–SOD-1–Catalase Conjugate Preserves Brain Endothelial Barrier Integrity Following Exposure to H_2_O_2_

Neuropathology associated with TBI is multifaceted, including activated immune cells, free radicals, proinflammatory cytokines, and astrocytes in the CNS. These processes in the acute and chronic periods can lead to endothelial cell dysfunction, which results in a damaged and leaky BBB [14,33]. A key property of the brain endothelial cells that comprise the BBB is the formation of a highly selective physical barrier that keeps the CNS microenvironment tightly regulated [34,35,36]. Quickly following the primary injury in TBI, molecular mechanisms responsible for the secondary damage quickly spread beyond the localized impacted areas. Thus, vascular protection at the level of the BBB may offer a means to prevent disruption to neuronal function. b.End3 cells expresses endothelial tight junction proteins such as zonula occludens 1 (ZO-1), typically seen in human brain endothelial cells. Figure 3A shows an example of the key tight-junction protein, zona occludens 1 or ZO-1 expression in b.End3 cells. Expression of ZO-1 is continuous and localized at the cell borders, indicating the formation of an intact barrier. To look at bEnd.3 endothelial barrier functional integrity after insult with H_2_O_2_, a small-molecular-weight fluorescence tracer (3 kDa FITC–dextran) was used to evaluate paracellular permeability (“BBB leakiness”). Figure 3 shows that at early (1 h) time points, 500 μM of H_2_O_2_ increased the leakiness of the barrier by over threefold (*p* = 0.0005) compared to wells with only FITC–dextran present. In wells treated with 500 μM of H_2_O_2_ followed by 100 ng/mL of dual conjugates, the amount of FITC–dextran crossing the endothelial barrier was greatly reduced: by over twofold compared to the 500 μM of H_2_O_2_ at the early time point (*p* = 0.0026), and 1.5-fold at the late time point (*p* < 0.0001) as shown in Figure 3.

### 3.3. Mechanical Stretch of Brain Endothelial Monolayer Induces Strain-Associated Injury That Is Rescued by Anti-ICAM-1–SOD-1–Catalase Dual Conjugates

A novel biomimetic model of TBI-like mechanical insult was used on brain endothelial cells to trigger the key endothelial changes that occur following a neurotrauma. These changes include the well-known mitochondrial damage and dysfunction of all the cell types within the CNS after a TBI [37,38]. In this regard, the primary impact injury causes physical deformation that can disrupt or destroy the complexes of the electron transport chain [39], whereas secondary injury processes can lead to excessive ROS production that overwhelms mitochondrial antioxidant defenses [10,14,17,40,41]. Thus, experiments were developed in this strain device platform for conducting in vitro “TBI-like” injury phenotypes and to assess if the dual conjugate could help prevent mitochondrial dysfunction after mechanical stretch injury. Figure 4A shows a schematic outlining the important steps taken for combining TBI-like injury, treatment of the dual conjugate, and measurement of cellular bioenergetics. Briefly, cells were grown to confluence on flat-patterned stretch membranes and subjected to a 4 mm stretch (20% strain) with a dwell time and relaxation. Treatments or sham was then administered and after the cells were transferred from the stretch membranes to Seahorse 96-well plates. After 1–2 days of growth, the cells were then subjected to a Seahorse Mito stress test.

Figure 4B shows that membranes that were unstretched and membranes that had received a 20% strain. After stretching, cells are no longer confluent and microtears could be visualized. After cells were grown on the surfaces stretched, they were tested using the Mito Stress Test. Mito Stress Test results on cells that had been stretched on flat surfaces were in line with cells that are alive but have no metabolic reserve when compared to the unstretched cells (Figure 4C). Cells from the stretched orthogonal membrane had reduced OCR, but not to the same degree as the flat membranes (Figure 4C). Therefore, the flat membranes provided the ideal testing platform to see if the anti-ICAM-1–SOD-1–catalase dual conjugates could rescue mitochondrial function in these mechanical injured cells.

Figure 4D shows the results of brain endothelial cells subjected to a maximal stretch injury and then treated with anti-ICAM-1–SOD-1–catalase dual conjugates for 1 h. Cells that were subjected to a stretch injury only had little to no metabolic reserve as expected; however, those treated with the dual conjugates displayed a near full recovery of their mitochondrial capacity. While the treated cells did not reach the same oxygen consumption rate as the sham group (cells seeded on stretch membranes and subjected to the same procedures except the stretch injury), they responded to stress nearly as efficiently. Figure 4E shows the ECAR of brain endothelial cells from the same experiment, and the stretch-injured group shows a sharp increase in proton excretion relative to the dual conjugate-treated and untreated groups, indicating more damaged cells in the former.

### 3.4. Anti-ICAM-1–SOD-1–Catalase Reduces Intracellular Superoxide Content in Cells Treated with H_2_O_2_, SIN-1, and KO_2_

To test the hypothesis that the membrane-bound dual conjugates could help lower intracellular superoxide content, direct fluorescence visualization of this ROS was employed. Superoxide is a major ROS that occurs endogenously within cells, but its production greatly increases after a TBI, exacerbating secondary injury mechanisms. Other major contributors of oxidative stress after a TBI are H_2_O_2_ as well as peroxynitrite (ONOO-), which is derived from SIN-1 breakdown products and superoxide radicals. Therefore, the following results provide an overview of how the dual conjugates handle increased ROS/RNS loads that are present in high concentrations following TBI. Figure 5A shows a selection of images from these experiments and Figure 5B shows the quantitative results. H_2_O_2_ increased presence of superoxide by over fivefold compared to the untreated cells (Figure 5). Cells that were treated with the dual conjugates had significantly reduced superoxide. The unconjugated mixture also reduced superoxide levels, but less efficiently than the dual conjugate (Figure 5).

### 3.5. Detection of Dual Conjugates Internalized in Stimulated Brain Endothelial Cells

To determine whether the actions of the dual conjugate are restricted to binding to the cell surface or whether the dual conjugate was internalized within the cell following binding to ICAM-1, confocal imaging studies were performed. bEnd.3 cells were seeded and allowed to grow to confluence and then stimulated for 4 h with 50 ng/mL of bacterial lipopolysaccharides (LPS) from *Escherichia coli* (O127:B8). Z-stacks were collected from various monolayers immunostained with antibodies against ICAM-1 and bovine SOD-1 (using the 2F5 antibody clone), which specifically detects the SOD-1 form present in the conjugate. Cells were also counterstained with DAPI to identify the cells containing the conjugate. The results in Figure 6 show immunopositive detection of ICAM-1 in the activated endothelial cells (green). Anti-SOD-1 detection showed the dual conjugate was internalized in some cells, but not others. Internalization was determined by evaluating the volumetric rendering of the z-stack where the top, midpoint and bottom of the cell were identified. The representative images show the midpoint level of the optical slices taken. The results suggest that the dual conjugate likely reduces oxidative stress extracellularly but importantly also within the cell, which supports the advantageous effects seen in the bioenergetic assays.

### 3.6. Histological Indices of CCI-TBI Neuropathology Are Attenuated by Administration of Dual-Conjugate (Anti-ICAM-1–SOD-1–Catalase) Intervention

Immunohistochemistry analysis was performed 48 h after impact for known markers of TBI pathology in the CCI mouse model of TBI. NeuN is a neuronal specific nuclear antigen and was used to determine viability of mouse neurons following impact injury and treatment group exposure. In live neurons, intranuclear NeuN is bound to the nuclear matrix, which disappears in damaged or dead neurons. NeuN immunoreactivity can also present as decreased staining in compromised/dying neurons during hypoxia and neurotrauma. Figure 7A shows representative images for the four groups stained for NeuN. Cells were counted in the treatment groups and compared to sham mice (Figure 7B). While NeuN staining was reduced in all impact groups, the group treated with dual conjugate showed significantly greater numbers of surviving neurons compared to the impact-only group (Figure 7A). GFAP was used as a marker for astrocytes that were actively responding to the head impact injury, indicating an inflammatory state [42]. The naïve mice showed immunopositivity for the astrocyte-specific protein marker GFAP. Astrocytes that express GFAP quickly upregulate it when astrocytes become activated and present as gliosis. Resting astrocytes are dispersed evenly throughout the tissue and the processes are thin and project to all directions. In the TBI condition, astrocytes upregulate GFAP expression and begin generating glial scars (gliosis) in areas of pronounced damage (Figure 7A). In the dual conjugate-treated animals, GFAP staining indicates a lower level of gliosis in the impacted region. Iba-1 is a sensitive protein marker that aids in the identification of resting and activated microglia. Immunopositive Iba-1 in the naïve group shows resting microglia as ramified, with long processes. Conversely, Iba-1 staining in the CCI-TBI group shows clear changes that depict activated microglia in the form of less ramification and more intensely labeled with Iba-1. Figure 7A shows that Iba-1 was increased in all groups relative to naïve mice; however, in the dual conjugate-treated group, it was significantly reduced compared to the CCI-TBI (Figure 7A).

In Figure 7B, cell count per area of NeuN+ neurons was measured and compared between the groups. A decrease of sixfold in NeuN was observed in the CCI-TBI and unconjugated mixture groups compared to naïve, while only a twofold decrease was seen in the conjugate-treated group. Live neuronal detection was significantly higher with targeted therapy than without, threefold higher, indicating neuronal preservation and reduced neuronal death after CCI-TBI (Figure 7B). GFAP staining was assessed by optical density per area and can be regarded as a sensitive and reliable marker of reactive astrocytes responding to injury in the CNS. GFAP staining of the naïve condition shows little baseline expression of GFAP in cortical levels. Following CCI-TBI, however, GFAP expression level is greatly increased, with an approximately fourfold increase in optical density (Figure 7B). GFAP expression level is reduced toward the naïve group upon administration of the dual conjugate (twofold increase). Quantification of Iba-1 positive microglia was performed for the activated status of the cell. Thus, the number of amoeboid and intensely labeled microglia was counted per area in the area of impact (or at the same anatomical coordinates in the naïve group) and compared between the groups. The number of activated microglia per area was significantly increased by nearly sixfold following CCI-TBI, and injury with targeted conjugate therapy managed to keep the increase to threefold only.

Together, these results suggest a neuroprotective and antineuroinflammatory effect from the administration of targeted dual antioxidant conjugates following CCI-TBI.

## 4. Discussion

According to the CDC’s latest reports from 2021, there were 69,413 TBI-related deaths, and nearly 2500 of those were children or adolescents. Children and teenagers are particularly vulnerable to head injuries, as the sequelae of TBI disrupt normal neurocognitive and behavioral development [43]. While public awareness of concussions, TBI, and their dangers has grown significantly, there are still no effective clinical options to treat patients of all ages [44]. Our group’s work is a step in the direction of filling this therapeutic gap for patients suffering from TBI.

It is known that a TBI will cause damage to the BBB, leading to endothelial cell activation, inflammatory responses and potentially a BBB breach, allowing toxic substances to enter the CNS parenchyma [45,46]. Previously, we explored attenuating the oxidative stress response following TBI in a mouse model using a novel targeted single antioxidant enzyme therapy [23]. Also, our team pioneered the strategy of vascular immunotargeting of drugs and drug carriers to surface determinants of the endothelial cells [47,48]. Studies of targeting antioxidant agents to PECAM-1, stable and highly expressed pan-endothelial cell adhesion molecule, provided the proof of concept for precise antioxidant interventions detoxifying endothelial ROS [49]. More recently, we expanded the arsenal of the targeting of SOD or catalase to ICAM-1, providing advantages of enhanced delivery to pathologically activated endothelial cells and relatively well-controlled intracellular delivery of the antioxidant cargoes [50,51,52,53]. The current study builds on this work by now using two antioxidant enzymes (rather than a single one, previously tested) conjugated with ICAM-1. Here, we demonstrate that this novel dual-conjugate construct (anti-ICAM-1–SOD-1–catalase) protects brain endothelial cells from the deleterious effects of TBI by counteracting BBB leakiness and the increased levels of free radicals while promoting mitochondrial health.

Mitochondria are damaged in the primary injury of TBI through excessive stretching forces as well as the secondary injury phase by both nonmitochondrial and intramitochondrial biochemical cascades. Outside the mitochondria, excitotoxicity leads to increased intracellular calcium ion levels, which in turn activates pro-oxidative enzymes such as xanthine oxidase and nitric oxide synthase. Within the mitochondria, there are numerous biochemical pathways and enzymes, including at the electron transport chain or ETC, which generate large amounts of ROS. These systems produce free radicals such as superoxide and ROS like H_2_O_2_ that under physiological conditions are reduced by endogenous antioxidant enzymes. Following TBI, however, the natural antioxidative protective systems quickly become overwhelmed by the massive increase in ROS, which subsequently leads to intracellular damage through cell membrane lipid peroxidation, DNA damage, and inhibition of the ETC function. Two major imbalances then follow: one is in the redox status of the cell, while the other corresponds to the energetics within the cell that can ultimately lead to cell death unless corrected. The cell has a high energetic demand to repair the damage caused by TBI that is hampered, as damaged mitochondria are unable to provide the ATP needed for repair mechanisms to fully function. Therefore, we investigated the effects of direct ROS exposure to monolayers of brain endothelial cells along with the anti-ICAM-1–SOD-1–catalase dual conjugates. We sought to determine the level of mitochondrial damage induced by H_2_O_2_ as measured by ATP assays and the cellular bioenergetics measuring platform of Agilent’s Seahorse Mito Stress assays.

H_2_O_2_ causes the cells to enter a highly glycolytic state (Figure 2A), shifting ATP production away from the more efficient process of aerobic respiration. Additionally, the H_2_O_2_ resulted in a drastically reduced ability of cells to respond to mitochondrial stress in the form of an uncoupling agent during the Mito Stress Test (Figure 2B). A metabolic shift towards glycolytic ATP production as well as an inability to accommodate the increased energy demands are pathological hallmarks following TBI. Figure 2B shows that the dual conjugate was able to rescue these deficits in a dose-dependent fashion, suggesting that reducing H_2_O_2_ following TBI via targeted antioxidant delivery (Figure 2C) is advantageous against untargeted antioxidant enzyme delivery to brain endothelial cells.

Next, the ability for the dual conjugates to provide protection in the context of a mechanical stretch injury applied to the cells was evaluated. Of note, the strain was intended to simulate similar stretching and shear forces experienced by cells during a TBI [54]. To our knowledge, this is the first example of combining the use of a stretch device to mimic TBI-associated forces with the Seahorse technological platform to determine cellular bioenergetics (Figure 4A). Figure 4D shows that the stretch-injured cells follow the abovementioned patterns of TBI-damaged cells: an increased reliance on glycolysis for ATP production as well as the inability to properly respond to stress. These cells display little to no metabolic reserve postinsult, posing a key limitation for brain endothelial cell recovery from injury. Administration of the dual conjugate provided the cells with greater resilience (reserve) following stretch injury. As such, they had increased metabolic reserve, as observed by OCR readings of patterning closer to measurements seen in the sham condition. ECAR in cells treated with the dual conjugate also followed the sham condition, whereas the stretch-injured cells showed a rapid increase in the rate of glycolysis (Figure 4D). These results suggest that the targeted delivery of the dual conjugates to injured brain endothelial cells conveys near-immediate metabolic benefits, as measured by mitochondrial health, allowing faster cellular recovery.

Another hallmark of TBI is disruption of BBB function and stability at the level of brain microvascular endothelial cells [55,56,57]. Mechanical or chemical injury can cause a breakdown of the BBB and the tightness between cells of the monolayer [58,59]. Therefore, we set forth to determine how H_2_O_2_ can alter the permeability of the BBB and if the dual conjugates could have a healing effect in the context of such an injury. Low-molecular-weight FITC–dextran (4 kD) was used to measure BBB permeability after injury at both early (1 h) and late (3 h) time points in comparison to groups treated with the dual conjugates. Our results suggest that H_2_O_2_ transiently increases the leakiness of the endothelial cell barrier, whereas treatment with the dual conjugates prevents the increase in paracellular permeability to a level approaching control groups. The results point to a direct benefit on barrier integrity when free radical damage is minimized following the use of this targeted intervention.

TBI induces an immediate production of superoxide, which begins a cascade of damaging oxidative stress, starting with the breakdown of superoxide into oxygen and H_2_O_2_. Additionally, superoxide can react with nitric oxide (NO), forming the highly oxidative peroxynitrite molecule, which can further cellular damage. Since our previous experiments explored the direct influence of H_2_O_2_ on dual conjugates or the indirect influence of the full superoxide cascade through mechanical stretch injury, we conducted experiments to directly measure the ROS reduction of the dual conjugates when exposed to superoxide, H_2_O_2_, or SIN-1, which can form both superoxide and peroxynitrite [60]. Administration of H_2_O_2_, superoxide via KO_2_ or SIN-1 resulted in clearly evident increases in ROS measured by superoxide MitoSOX red fluorescence in this particular set of experiments. Adding the anti-ICAM-1–SOD-1–catalase dual conjugate after administering the above chemical insults greatly reduced the ROS signal, whereas administration of the unconjugated mixture control group did not have as drastic an effect (Figure 5). These results suggest that the targeted dual conjugates can have a beneficial effect on brain endothelial cells experiencing oxidative stress, such as that seen in TBI. It stands to reason that the anti-ICAM-1 antibody allows for the SOD-1 and catalase enzymes to be brought in close proximity to the cell, which helps accelerate the breakdown of superoxide and subsequently H_2_O_2_.

Although the abovementioned studies provided a better understanding of the benefits conveyed by the dual conjugates at the cellular level, we wondered whether similar outcomes would be observed in vivo. Therefore, the studies here also attempted to gather information as to the therapeutic effectiveness of the dual conjugate in vivo via a controlled cortical impact mouse model of TBI. Established neuropathological indices of TBI were used to conduct this study, NeuN to monitor cortical neuron loss after injury along with GFAP and Iba-1 for astrocytic and microglial activation, respectively. These markers provide similar tissue pathology in both mouse models of TBI and in humans who have experienced neurotrauma [61]. Consistent with our previous published work along with other groups, in moderate CCI-TBI, affected mice displayed decreased NeuN levels concurrent with increased markers of neuroinflammation—GFAP and Iba-1. In TBI patients, the resulting chronic neuroinflammation can be extremely debilitating, leading to cognitive and motor deficits along with other comorbidities, such as substance abuse [62]. Targeting neuroinflammation in the acute phase of TBI through reducing the overabundance of ROS could help reduce persistent activation of astrocytes and microglia (Figure 6). Significantly, administration of anti-ICAM-1–SOD-1–catalase 4 h after CCI-TBI resulted in increased neuronal survival and decrease in glial activation, whereas the unconjugated mixture of anti-ICAM-1, SOD-1, and catalase did not convey the same beneficial effect (Figure 6). These results provide strong evidence that targeted antioxidant enzyme therapy to the brain vasculature is highly effective for rendering neuroprotection following a brain injury.

The results presented in this report provide strong proof of concept that two potent antioxidant enzymes can be covalently fused to a targeting antibody in a single construct that is both biologically active and targets inflamed brain endothelial cells. It remains to be tested, but it is thought that such an approach of minimizing trauma-associated neuroinflammation could also extend to other pathological conditions in which the cerebral vasculature is activated and the BBB is compromised.

## Figures and Tables

**Figure 1 bioengineering-11-00200-f001:**
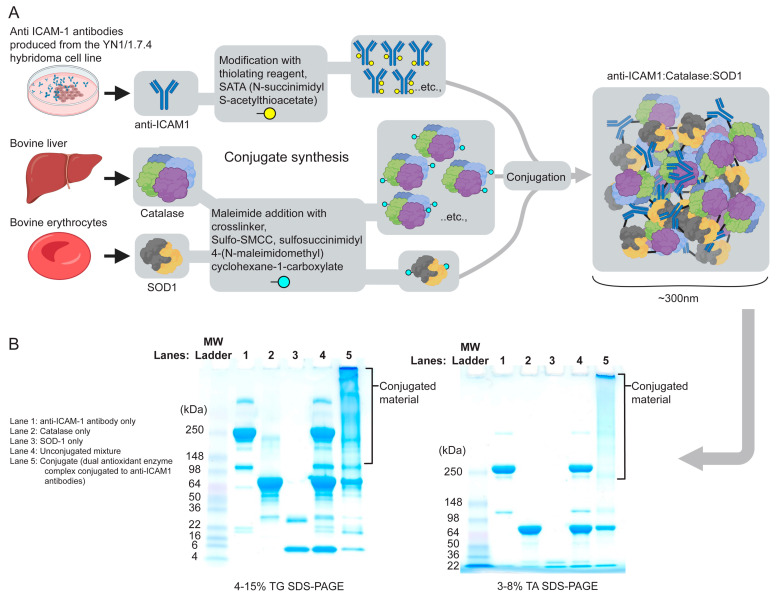
Development of engineered dual antioxidant enzyme–antibody conjugate complexes. (**A**) Complexes are generated by “click” chemistry. First, protected sulfhydryls were introduced to the ICAM-1 antibody using N-succinimidyl-S-acetylthioacetate (SATA). Additionally, maleimide groups were added onto the antioxidant enzymes catalase and SOD-1 by the heterobifunctional cross-linker succinimidyl 4-[N-maleimidomethyl]cyclohexane-1-carboxylate (SMCC). Sulfhydryls were deprotected and a conjugation reaction was initiated at a ratio of 2:1:1 (antibody:catalase:SOD1). Unreacted components were removed using Zeba Spin Desalting purification columns. (**B**) Molecular weights of conjugate complexes were analyzed by SDS-PAGE. Comparisons were made with anti-ICAM-1 antibody only (lane 1), catalase only (lane 2), SOD-1 only (lane 3), unconjugated mixture (lane 4), and conjugate complex (lane 5).

**Figure 2 bioengineering-11-00200-f002:**
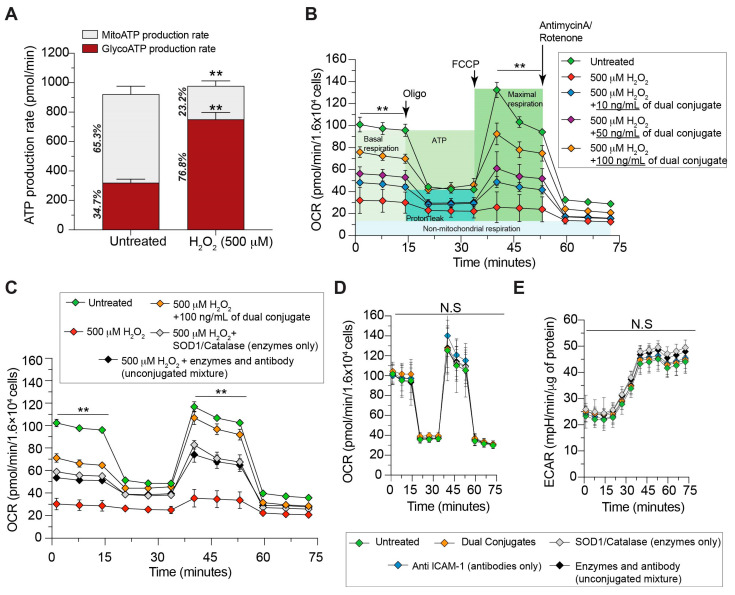
Dual conjugates prevent compromised mitochondrial respiration in brain endothelial cells during oxidative stress insult. Cells were seeded at 16,000 per well and grown until confluence. The cellular oxygen consumption rate (OCR) bioenergetic profile under the indicated conditions was measured in real time using the Seahorse XFe96 Extracellular Flux Analyzer. (**A**) The effect of the experimental dose of H_2_O_2_ (500 μM) on cellular ATP production source, showing a dramatic shift to glycolytic metabolism following insult; (**B**) dose-dependent assay of dual conjugate (100 ng/mL, 50 ng/mL, 10 ng/mL) in response to 500 μM of H_2_O_2_; (**C**) same schema as panel, B except 100 ng/mL of dual conjugate is compared to equal doses of control groups: free SOD-1 and catalase or an unconjugated mixture of SOD-1, catalase, and anti-ICAM-1; (**D**) OCR and (**E**) extracellular acidification rate (ECAR) measurements to determine any basal toxicity of dual conjugates. Vehicle, dual conjugates, or previously mentioned control groups in addition to an anti-ICAM-1 group alone were exposed to cells for 1 h. For (**A**), means ± SEM, n = 6, ** *p* < 0.001. For (**B**–**E**), two-way ANOVA with Dunnett’s test for multiple comparisons was used. N.S = not significant. Baseline and stress-testing phase (FCCP injection) OCR values for (**B**) (*p* < 0.0001) and (**C**) (*p* < 0.0001) were significantly higher for both the untreated and 100 ng/mL dual-conjugate group compared to different conjugate doses (**B**) or controls (**C**).

**Figure 3 bioengineering-11-00200-f003:**
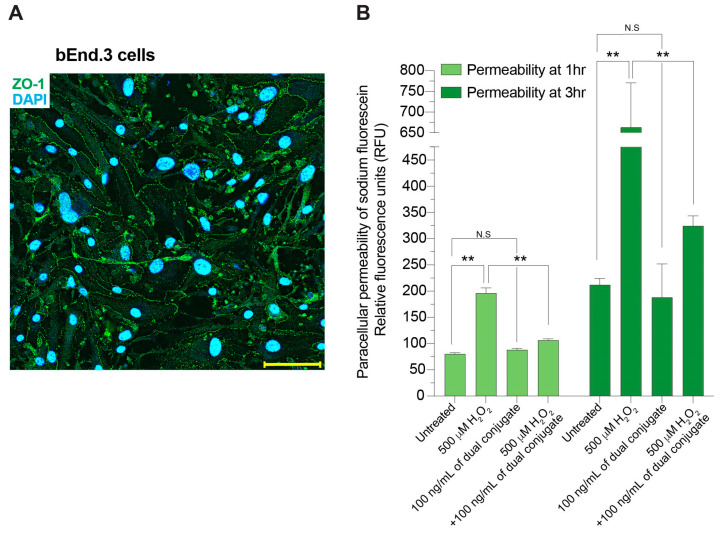
Engineered dual conjugates attenuate H_2_O_2_-induced increases in brain endothelial cell permeability. (**A**) b.End3 cells were examined by immunofluorescence staining for the expression of the tight-junction protein zonula occludens 1 (ZO-1). ZO-1 (green) was localized at the cell borders and DAPI (blue) stained the cell nuclei. Scale bar equals 10 microns. (**B**) bEnd.3 cells were grown on transwell membranes until confluence. Cells were then treated with 500 μM H_2_O_2_ with or without 100 ng/mL of the dual conjugates. Permeability was assessed at 1 h and 3 h using a 3 kDa fluorescent weight dextran that was applied to the top of the well and measured from the bottom chamber. Relative fluorescence units were determined using a SpectraMax M5e. Treatment with the conjugates reduced permeability changes from exposure to H_2_O_2_. ** *p* < 0.001, n = 3 replicates each condition. N.S = not significant.

**Figure 4 bioengineering-11-00200-f004:**
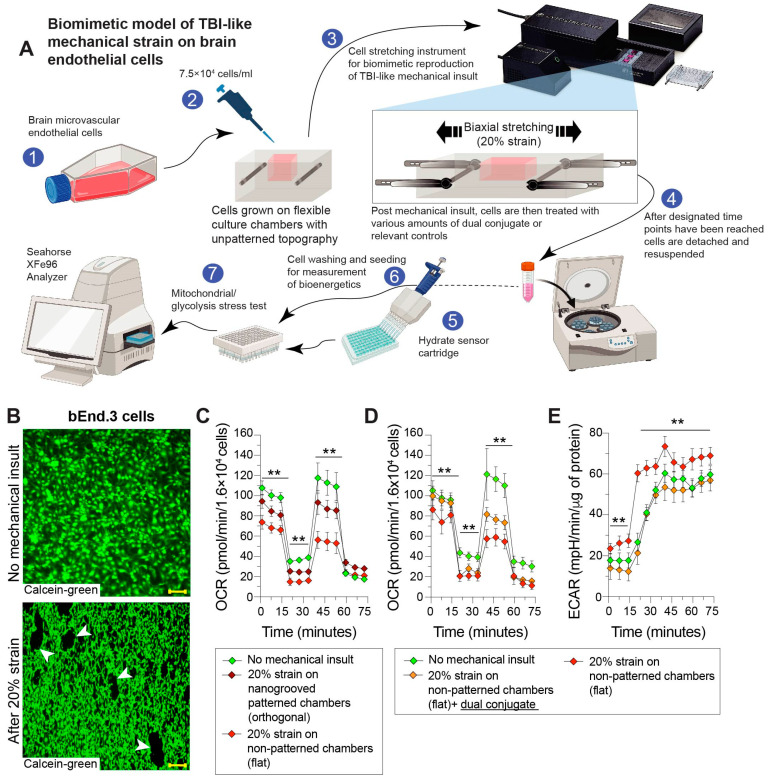
Dual conjugates prevent mechanical stretch-induced damage to brain endothelial cells. Cells were seeded at a density of 75,000 per stretch membrane and grown to confluence. (**A**) Workflow used to develop this model and schema used in the rest of the panels; (**B**) fluorescent image showing microtears in the endothelial cell monolayer following a stretch with 20% strain (bottom image) compared with a sham group (top image); scale bar 200 microns; (**C**) initial tests to determine which membrane surface, flat or with nanogrooves orthogonal to the direction of stretch, induced more damage to endothelial cells; (**D**) comparison of OCR between sham, stretch-injured, and injury + treatment groups; (**E**) same experiment and cells as (**D**), measurement of ECAR values. Statistical significance is omitted for clarity. For (**C**–**E**), two-way ANOVA with Tukey’s test for multiple comparisons was used. Baseline and stress-testing phase (FCCP injection) OCR values for (**C**,**D**) are significantly different for all groups (** *p* < 0.001).

**Figure 5 bioengineering-11-00200-f005:**
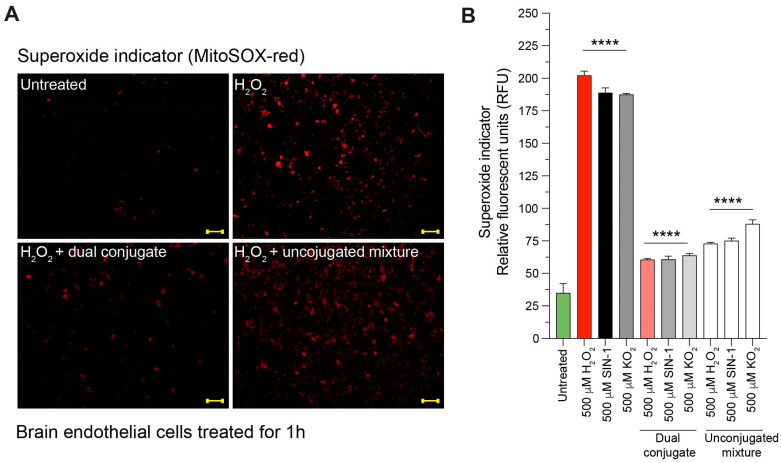
Dual conjugates reduce superoxide production following free radical insult from H_2_O_2_, SIN-1, or KO_2_. Cells were seeded at 55,000 per well (24-well plate) and grown to confluence. (**A**) Comparison images for groups exposed to 500 μM H_2_O_2_ (other treatment groups not shown). The lower-left image shows cells treated with dual conjugates had greatly reduced concentrations of superoxide (red) compared with the insult only or insult + unconjugated mixture images on the right side. Scale bar 200 microns. (**B**) Fluorescence intensity of MitoSOX red superoxide indicator comparison between all experimental groups, including cells treated with SIN-1 (a superoxide and peroxynitrite producer) and KO_2_ (a potent superoxide producer). For (**B**), two-way ANOVA with Tukey’s test for multiple comparisons was used; **** *p* < 0.0001.

**Figure 6 bioengineering-11-00200-f006:**
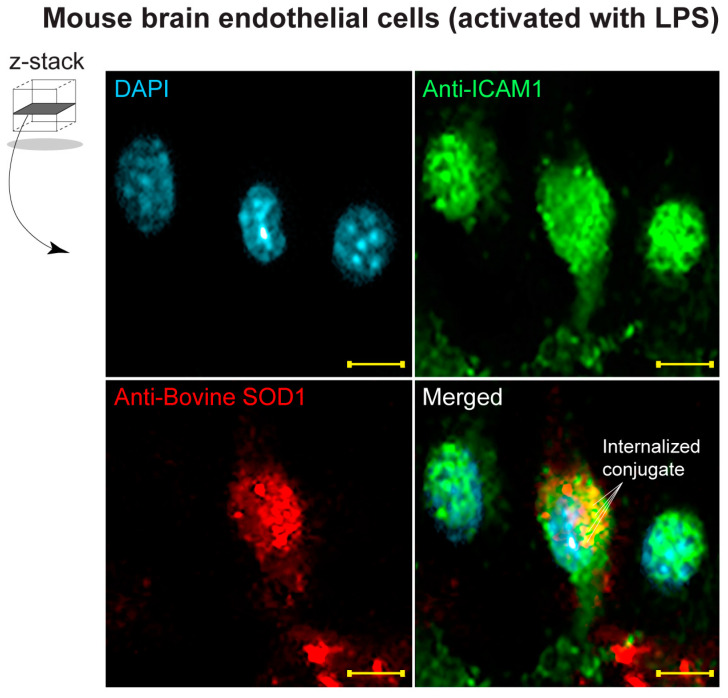
Internalization of dual-conjugate constructs within brain endothelial cells. Optical slices from confocal imaging. Z-stacks identified the top, midpoint and bottom aspects of the cells. Brain endothelial cells (bEnd.3) were stimulated with 50 ng/mL of bacterial LPS for 4 h and then fixed and immunostained with anti-ICAM-1 (green), anti-bovine SOD-1 (red) and nuclear counterstained (DAPI). Labels indicate internalized dual conjugates. Scale bar = 10 microns.

**Figure 7 bioengineering-11-00200-f007:**
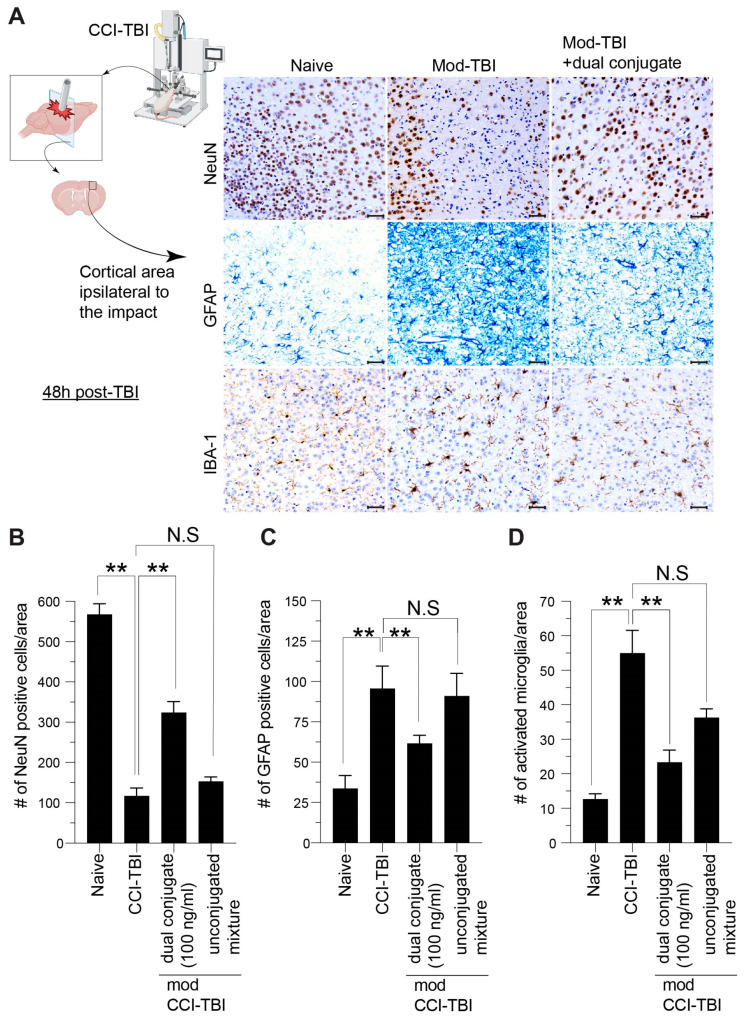
Representative images and quantitative evaluation of post-TBI neuropathological indices for neuronal cell death, astrocyte and microglial activation in animals treated with or without dual conjugates. Mice were placed in the following groups prior to brain harvesting and immunohistochemistry: naïve (craniotomy only but without impact), moderate CCI-TBI and moderate CC-TBI + 100 ng/kg of dual conjugate administered i.v. (**A**) NeuN staining identifies viable neurons within the cerebral cortex and lack of DAB+ staining for NeuN demonstrates the substantial neuronal loss at 48 h following CCI-TBI. The combination of mechanical strain, oxidative stress and inflammatory responses inevitably results in cell death. However, a rescue of viable neurons can be observed with the presence of dual conjugate administered soon after the TBI was induced. Astrocyte activation was determined by GFAP expression. CCI-TBI shows significant increase at 48 h following CCI-TBI, which is attenuated with the presence of the dual conjugate. Similarly, evaluation of activated microglial by phenotype and augmentation of expression of IBA-1 is clearly seen in the TBI animal. Dual conjugates also reduced the degree of microgliosis. Scale bar 50 microns (**B**) Bar graphs show the imaging quantification of the neuropathological indices from the groups outlined above. Data are presented as means ± SEM. For (**B**–**D**), one-way ANOVA with Dunnett’s test for multiple comparisons was used. ** *p* < 0.001, N.S = not significant.

## Data Availability

Data are available upon reasonable request.

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
