# Peer review of "Engineered Dual Antioxidant Enzyme Complexes Targeting ICAM-1 on Brain Endothelium Reduce Brain Injury-Associated Neuroinflammation"

_bioengineering, 2024, doi:10.3390/bioengineering11030200_

Round 1

Reviewer 1 Report (Previous Reviewer 1)

Comments and Suggestions for Authors

Dear authors, the manuscript presents an objective methodology that is compatible with the results obtained. 

Author Response

We appreciate the Reviewer's careful evaluation of our manuscript.

Reviewer 2 Report (Previous Reviewer 2)

Comments and Suggestions for Authors

The best thing about this article is that it discusses a new molecule –its synthesis, characterization, and biological activity. The authors must be appreciated for coming up with a good plan. However, there are still a few areas that are not clear and need to be explained

1.  How come the MW of an antibody is 250 KDa? A Full-length antibody composed of heavy chain and light chain will be ~150 KDa (if it is monomeric, like IgG) or ~300 KDa (If it is dimeric, like IgA). Further, due to the reduction of the intramolecular –SH group, the heavy chain and light chain must separate in SDS-PAGE to show MW of 50 and 25 KDa. This antibody MW in the gel is very difficult to explain. Please explain.

2. Based on the SDS-PAGE analysis it seems that the percentage of conjugation is less which is likely. But how do you purify this conjugate from others? A 7 KDa MWCO is not going to remove the free antibody and other enzymes. Have you tried gel-filtration Chromatography to purify the conjugate from others?

3. It is likely that less SOD activity is obtained because the amount of loaded conjugate can NOT be measured?

4. Nowhere it is mentioned what is the actual MW of the conjugate. It should be above 560 KDa based on the 2:1:1 conjugation efficiency. Have you tried to measure the actual MW of the compound because SDS-PAGE Data does not provide any idea about the conjugate MW?

5. A 6-7% SDS-PAGE must be run to see the actual MW of the conjugate only. Authors must understand for this kind of study the chemistry and characterization part is VERY important even if the biological activity may be a little compromised.

6. It is not mentioned where in the antibody (the amino acid residue) the conjugation chemistry is done. Although it was shown in the Figure 1 image that it is in the Fc region of the antibody, the actual location of modification should be mentioned

Comments on the Quality of English Language

only minor corrections required 

Author Response

Reviewer 3 Report (Previous Reviewer 3)

Comments and Suggestions for Authors

Scale bars can be added to some figures.

Author Response

We are grateful for the Reviewer's comments. As instructed, we have revised the scalebars in all the figures for improved visibility. Note, the dimensions in the figure legends. 

Round 2

Reviewer 2 Report (Previous Reviewer 2)

Comments and Suggestions for Authors Author's have earlier said that 2:1:1 ratio of conjugation. However, now they are saying in many places due to the Lysine residues ratio of the conjugation can  happen. Therefore, they must reframe the 1st sentence. The optimum conjugation ratio is 2:1:1 but other ratios of conjugation can also happen.  So actual MW of the complex will be a heterogeneous mixture of high MW conjugates of different ratios. Hence actual MW calculation will be difficult for the conjugated Products.  Authors are advised to Run a Gel filtration Chromatography with standards to identify the MW and size of the conjugate in the future. Also MS analysis in the future for this kind of compound.  The good thing is that the gel shows a complete conversion of antibodies and other molecules to the conjugate.  Authors are also advised to give the catalog no and company name they have used for this Antibody in the manuscript.   This will clear the doubt of antibody MW

Author Response

This manuscript is a resubmission of an earlier submission. The following is a list of the peer review reports and author responses from that submission.

Round 1

Reviewer 1 Report

Comments and Suggestions for Authors

Dear authors,

The manuscript describes the use of anti-ICAM-1 antibodies conjugated to CAT and SOD-1 to reduce the effects of oxidative stress on neuroinflammation.

The text is well written. The methodology is sufficiently detailed, and the results obtained are compatible with them. And I have just two questions for the authors.

1) In "in vivo" experiments, only 3 animals/treatment were used. The authors should justify the use of this number of animals, since the experiment was not repeated. In general, the number of animals used in similar experiments is greater, and ideally the experiment should be repeated more times, as was done in "in vitro" studies in this same study.

2) It is not possible to analyze figure 3. Item 3B is not visible.

Reviewer 2 Report

Comments and Suggestions for Authors

Reviewer’s comments for Bioengineering 2645717

This work is really interesting because it talks about targeted delivery using antibody-conjugated drugs. However, there are a few areas where clarification is required to make the manuscript better.

1. Authors have used only the DLS method to justify the conjugated high MW product. That is NOT enough. Please provide native/SDS-PAGE data to show the shift in the MW after conjugation. If possible, you can provide proteomics data to provide the same.

2. How do the authors know that there is a 2:1:1 conjugation? What experiment they have done to make sure there is no heterogeneous mixture of conjugated products?

3.  Have you measured the stability of the conjugate at different conditions (temperature, pH, salt, etc.)? Whether the SOD and Catalase is stable in the conjugate?

4. Although the authors have done similar chemistry earlier, they must indicate the position of -SH and -NH2 groups that are protected or used. Further, how they measure the quality of maleimide-conjugated proteins and their % conversion. Please provide data.

5. Please provide a brief reasoning about why ICAM-based targeting was done here. ICAM-1 was found to be very well expressed in endothelial cells and immune cells. How this antibody-based targeting will be effective specifically against TBI without affecting other cells?

6. Provide a phase contrast image of cells, in Figure 3.

7. Fig 5B data shows there is no significant difference in superoxide ions reduction in the presence and absence of SOD + Catalase in the conjugate. What is the explanation?

8. Author’s have done immunofluorescence imaging to prove that ICAM and SOD are co-targeted. Since they have the antibody they can try to detect internalized SOD by western blot.

9. Authors have used the proteins and antibodies from commercial sources to design the conjugate. There is a significant cost factor there when we consider large-scale possible applications. Please discuss this matter shortly in the discussion part. How cost-effective this delivery will be compared to standard delivery for TBI?

Reviewer 3 Report

Comments and Suggestions for Authors

In this manuscript, “Engineered dual antioxidant enzyme complexes targeting ICAM-1 on brain endothelium reduces brain injury associated neuroinflammation” by Leonard et al. explores the notion that attenuating oxidative stress at the vasculature after TBI may result in improved BBB integrity and neuroprotection. This work is well designed, but without agreed results. Therefore, I would suggest that authors may take a major revision before publication. Here are the comments and suggestions:

1.        In Fig. 1, the ratio of antibody: catalase: SOD1 seems incorrect.

2.        The response for untreated seems disagreed in Figs. 2B, 2C and 2D.

3.        The response for “no mechanical insult” seems disagreed in Figs. 4C and 4D.

4.        In Fig. 7, please explain if increase the dual conjugate at higher concentration than 100 ng/mL could have the better results?